# A Structural Potential of Rare Trinucleotide Repeat Tracts in RNA

**DOI:** 10.3390/ijms23105850

**Published:** 2022-05-23

**Authors:** Dorota Magner, Rafal Nowak, Elzbieta Lenartowicz Onyekaa, Anna Pasternak, Ryszard Kierzek

**Affiliations:** Institute of Bioorganic Chemistry, Polish Academy of Sciences, Noskowskiego 12/14, 61-704 Poznan, Poland; dorotaw@ibch.poznan.pl (D.M.); rnowak@ibch.poznan.pl (R.N.); elenartowicz@ibch.poznan.pl (E.L.O.); apa@ibch.poznan.pl (A.P.)

**Keywords:** RNA repeats, RNA trinucleotide repeats interaction, RNA thermodynamics, RNA trinucleotide repeats secondary structure, *GABRA4* 3′UTR secondary structure, UAA-UUG RNA duplexes, *CHIC1* mRNA quadruplexes

## Abstract

Among types of trinucleotide repeats, there is some disproportion in the frequency of their occurrence in the human exome. This research presents new data describing the folding and thermodynamic stability of short, tandem RNA repeats of 23 types, focusing on the rare, yet poorly analyzed ones. UV-melting experiments included the presence of PEG or potassium and magnesium ions to determine their effect on the stability of RNA repeats structures. Rare repeats predominantly stayed single-stranded but had the potential for base pairing with other partially complementary repeat tracts. A coexistence of suitably complementary repeat types in a single RNA creates opportunities for interaction in the context of the secondary structure of RNA. We searched the human transcriptome for model RNAs in which different, particularly rare trinucleotide repeats coexist and selected the *GABRA4* and *CHIC1* RNAs to study intramolecular interactions between the repeat tracts that they contain. In vitro secondary structure probing results showed that the UAA and UUG repeat tracts, present in *GABRA4* 3′ UTR, form a double helix, which separates one of its structural domains. For the RNA *CHIC1* ORF fragment containing four short AGG repeat tracts and the CGU tract, we proved the formation of quadruplexes that blocked reverse transcription.

## 1. Introduction

The RNA structure determines the activity of transcripts and their interactions with other molecules in cells. The presence and role of short repeat sequences in RNA are puzzling in the context of the structure of the transcripts. RNA repeats are derived from microsatellite sequences, also known as short tandem repeats (STR). They contain motifs of 1–10 nucleotides in length that repeat 5–50 times [1]. Their localization is not random, not limited to specific regions of the transcript, and presumably related to a structurally determined function. In mRNAs, STRs-derived tracts are found in 5′ and 3′ untranslated regions (UTR) as well as introns and exons [2,3,4,5]. STRs may act as cis-regulatory elements affecting transcription, splicing, and translation [6,7]. It was observed that repeats in RNA undergo selective pressure because some types are overrepresented, while others are underrepresented, depending on the length of the tract or region of occurrence. For example, open reading frames (ORFs) contain tri- and hexanucleotide repeats more frequently than other sizes/types of microsatellite repeats, as a result of selective pressure at the level of protein function and codon usage [8,9]. Among STRs, trinucleotide repeats are strongly represented in RNA with an overrepresentation of CNG type, whereas the other types of STR such as AAU, AAC, or CUU are underrepresented [10]. CNG tracts are particularly well-studied repeats as they are a disease-causing factor in several human disorders with neurological symptoms such as X-fragile syndrome, myotonic dystrophy, Huntington’s disease, or different types of spinocerebellar ataxia. Most of the information on RNA repeats comes from studies on the above Trinucleotide Repeat Expansion Disorders (TRED), the course of which is determined by the number of trinucleotide repeats found in ORFs, as well as UTRs of several human genes.

In 2013, the Weeks group demonstrated intramolecular pairing of RNA regions rich in CAG and CCG repeats for the first exon of the huntingtin transcript, proving the highly structured character of this RNA fragment [11]. In the case of normal huntingtin, the CAG repeat region forms extensive base pairing with nucleotides flanking it and containing CCG repeats. In case of abnormal CAG expansion leading to huntingtin mutation, an extended tract of CAG repeats forms itself an additional, stable hairpin, without changing the original interactions of the CAG and CCG repeats. It is well documented that GC-rich trinucleotide repeats in RNA, especially the CNG ones, form stable secondary structures depending on the length of the tract and the type of repeated motif [12]. On the other side, many RNA repeats do not form any particular structure, but their presence in the human genome may suggest some yet-undetermined biological functions. Analysis of genes containing trinucleotide repeats (TNRs) indicated that 11% of them harbor more than one type of TNR, many of them in exons [10]. Prediction of the secondary structure of mRNAs containing various multiple TNR tracts indicates that those repeat tracts can base pair even when they are separated in mRNA structure.

Our research goal was to determine the folding and thermodynamic stability of pairing of various, predominantly rare RNA repeat types. We found that repeats which do not fold in secondary structure by themselves can pair with another similar type of repeat. Based on the analysis of the coexistence of rare repeats in a single transcript, we chose fragments of two RNAs: *GABRA4* (gamma-aminobutyric acid type A receptor subunit alpha4) and *CHIC1* (cysteine-rich hydrophobic domain protein 1), containing tandem repeats that may potentially interact. In both cases, we performed an attempt at secondary structure chemical mapping to determine the relationship between the present repeats and their effect on the secondary structure of the RNA molecule.

## 2. Results and Discussion

### 2.1. RNA Repeats Thermodynamics

Characteristic sequences, such as repeated motifs, may play a special role in the context of RNA secondary and tertiary structure. We analyzed 23 types of RNA repeats by determining their preferred secondary structure, as well as their potential for forming structural motifs with other RNA repeats. For the thermodynamic studies, model oligonucleotides were used, containing three repeats and respective terminal base-pairs. For selected repeats, oligonucleotides containing six RNA repeats were also analyzed. The UV-melting experiments included the following RNA repeats: CAA, CGA, GCU, CGU, GCC, GCG, GCA, AGG, UGG, UAG, UAA, CAU, CUA, CUU, CCU, CCA, UUA, AUG, GAA, UUG, AAG, CCUG, and AUUCU. Collected thermodynamic results (Appendix A) indicated that the analyzed RNA repeats can form a duplex, quadruplex, or stay single-stranded/unstructured. (Table 1). The lack of transitions observed for melting curves suggested a single-stranded structure of the RNA repeats.

Duplexes were formed by eight types of repeats: CAA, CGA, GCU, CGU, GCA, GCC, GCG, and CCUG (Figure 1). Oligonucleotides composed of CGA, GCC, CGU, and GCG repeats formed strong duplexes, presenting a two-state character of melting with increasing thermodynamic stabilities (−ΔG⁰_37_) of 4.22, 5.29, 5.84, and 7.29 kcal/mol, respectively. The observed destabilizing effect of a single mismatch increased in the following order: G-G, U-U, C-C, and A-A, which is consistent with previous reports [13]. The CAA, GCU, and CCUG repeats formed quite stable duplexes, but their melting was not two-state. This may be due to the predominance of non-canonical base pairs (mismatches) in the duplex. A duplex formation by CAA repeat type was not reported previously. In the case of three GCA repeats, we observed a characteristic melting curve containing two transitions that we initially associated with the presence of triplex, in which the lower transition reflects the dissociation of the third strand and the upper one the duplex unfolding. However, a melt performed in pH 5.5 buffer at 295 nm did not confirm the presence of the triplex. The observed transitions correspond to two duplex structures with different folds and stability (−ΔG⁰_37_) of 4.27 and 8.65 kcal/mol, respectively.

The oligonucleotides containing three consecutive AGG or UGG repeats formed quadruplexes with thermodynamic stability equal to −2.95 and −3.18 kcal/mol, respectively. For the remaining types of RNA repeats, UV-melting curves do not show transitions, which demonstrates the single-stranded character of melting. This large group of RNA repeats, that do not form any structure, includes UAG, UAA, CAU, CUA, CUU, CCU, CCA, UUA, AUG, GAA, UUG, AAG, and AUUCU repeats. However, for some of those repeats (CCA and UAG), a kind of very low Tm transition was observed. Detailed thermodynamic parameters of the analyzed RNA repeats are collected in Appendix A.

Increasing the number of repeats in the case of certain GC-rich RNA oligonucleotides promotes their folding into hairpins. A thermodynamic study of oligonucleotides containing six GCU or GCC repeats indicated the formation of a mixture of two hairpins in each case. The free energy of those hairpins was −1.87 and −3.27 (for GCU) or −2.60 and −3.03 kcal/mol (for GCC). The formation of various hairpins structure by oligonucleotides containing six CNG repeats was reported earlier [12,14]. Often, the structural dynamic of TNR hairpin structures is related to the formation of various sizes of hairpin loops (four vs. seven nucleotides) or slipperiness of the double-stranded hairpin stem [12]. Interestingly, the oligonucleotides, each carrying six CGA, GCG, and GCCU repeats in the same conditions, formed single-hairpin structures, and their thermodynamic stabilities (−ΔG⁰_37_) were 1.10, 2.14, and 0.61 kcal/mol, respectively. Conversely to the information listed above, an oligonucleotide containing six CGU repeats results in the simultaneous formation of duplex and hairpin structures with free energy equal to −9.46 and −1.50 kcal/mol, respectively.

Previously it was shown that RNA composed of 5–69 CUG repeats forms hairpins whose melting temperatures (Tm) are similar (ca. 75 °C), no matter the number of repeats [15]. Detailed thermodynamic studies of RNA formed by CNG repeats (N=A, C, G, and U) indicated that depending on the CNG repeats number, such oligonucleotides can form self-complementary duplexes (2–3 repeats), a mixture of duplexes and hairpins (4–5 repeats), and hairpins alone (six or more repeats). The transition between duplex and hairpin formation was gradual, and the contribution of the hairpin increased with the number of repeats and concentration of the oligonucleotide. Double-stranded fragments of such structures (duplex and hairpin stem) contained a single mismatch for every third pair and, consequently, their thermodynamic stability was low. Moreover, the structure and pattern of hydrogen bonding and stacking was demonstrated by the crystallographic approach for structures containing CAG, CUG, CGG, and CCG repeats [16,17,18,19,20]. Structural diversity of triplet repeat RNAs was also demonstrated by Sobczak et al. [12]. In this study, transcripts containing 17 repeats of the various sequences were used. Using chemical mapping and a UV-melting approach, it was shown that CAA, UUG, AAG, CUU, CCU, CCA, and UAA do not form any structure, whereas the repeats CAU, CUA, UUA, AUG, UAG, CGA, CGU, CUG, CAG, CGG, and CCG form hairpins with various thermodynamic stabilities. Long repeats of AGG and UGG form quadruplexes. Later, also using NMR, circular dichroism (CD), and mass spectrometry (ESI-MS) approaches, it was confirmed that RNA composed of four AGG or UGG repeats forms quadruplexes in a broad spectrum of cations, whereas CGG quadruplex is formed preferentially in presence of potassium cation [21].

### 2.2. The Effect of Molecular Crowding

UV-melting performed in buffer containing 100 mM sodium chloride only partially reflects intracellular conditions. Cells contain a large number of nucleic acids, proteins, carbohydrates, and the total concentration of biomolecules is estimated to be 50–400 g/L, depending on the type of cells and cellular compartments [22,23,24]. To reflect cellular-like conditions in thermodynamic studies, molecular crowders are tested. Among them, the most frequently used are polyethylene glycol (PEG) and dextrin. Recently, the Znosko group determined the nearest neighbor parameters for RNA duplexes under molecular crowding conditions [25]. The thermodynamic measurements were performed in a buffer containing 1 M sodium chloride and a 20% solution of PEG200, in which the total ions concentration was 258 g/L. They found that the PEG presence destabilizes (ΔΔG⁰_37_) self-complementary RNA duplexes (mostly six- and eight-mers) by ca. 1 kcal/mol. The presence of PEG changed the particular nearest-neighbor parameters for RNA duplexes in the range 0.03–0.17 kcal/mol; however, significant difference (0.54 kcal/mol) was related to the free energy of initiation of RNA duplex formation.

Therefore, we have also determined the thermodynamic stability of duplexes and quadruplexes formed by RNA repeats under molecular crowding conditions. In the case of duplexes, thermodynamic measurements in presence of 200 mM PEG400 (80 g/L) were performed only for the RNA repeats which in standard (PEG-free) melting buffer demonstrated cooperative (two-state) transition (Figure 1, Appendix A). Consequently, for RNA duplexes formed by CGA, CGU, GCC, and GCG repeats, PEG400 presence in melting buffer resulted in free-energy change of 0.15, 0.15, −0.02, and −0.10 kcal/mol, respectively, and these differences were within the range of experimental errors.

Oligonucleotide composed of three GCA repeats invariably presented two transitions of melting curves. UV-melting in buffer containing 200 mM PEG400 changed low- and high-temperature transition by 0.98 and −0.29 kcal/mol, respectively, reflecting destabilization of the weaker GCA duplex and stabilization of the GCA core duplex. For quadruplexes formed by AGG and UGG repeats, the presence of PEG in melting buffer changed their free energy by 0.43 and −0.28 kcal/mol, respectively.

The results showed that the presence of the additional 200 mM of PEG 400 in the UV-melting buffer, already containing 100 mM sodium chloride, did not influence RNA structures formed by particular repeats. Moreover, the contribution of PEG400 to thermodynamic stability was also low, usually within 10% of original free-energy values.

### 2.3. Inter-strand Interactions of Various RNA Repeat Tracts

More than half of the analyzed oligonucleotides formed by the RNA repeats remain single-stranded. These repeats are rich in adenosine, uridine, and cytidine. Among them, we selected pairs that could potentially form duplexes and determined their thermodynamic stability by UV melting in two types of buffers (Appendix A). Standard melting was performed in a buffer containing 1 M sodium chloride to provide an optimal temperature range for the melting curves of analyzed oligonucleotides. In addition, a physiological buffer containing 40 mM potassium chloride and 8 mM magnesium chloride was applied. We observed the formation of inter-repeat mismatched duplexes for eleven pairs of repeats (Figure 2).

The free energy of those duplexes ranged from −3.4 to −4.8 kcal/mol, and their melting temperatures did not exceed 30 °C. About half of the analyzed duplexes melted in a two-state manner, which means that no intermediate states between the single-stranded form and the duplex were observed in the solution. For A, U, and C-rich repeats, the formation of a duplex is dependent on the types of mismatches occurring next to conventional Watson–Crick base pairs. For such duplex-forming repeats, the canonical, stabilizing pairs A-U may be insufficient to maintain a helix that contains highly destabilizing mismatches, e.g., C-C or A-A. For duplexes formed by such types of repeats, the G-U Wooble pair’s stabilizing role that supports these duplexes is significant. We observed a stronger destabilization of these duplexes in the presence of magnesium and potassium ions in comparison to the standard buffer, and the destabilizing effect was about 1.5–2 kcal/mol (Appendix A). However, this trend was not a rule for repeats including G and C residues, for which the presence of magnesium and potassium ions stabilized duplexes formed by some repeat pairs. These observations suggest that single-stranded, AU-rich trinucleotide RNA repeats have a considerable potential for duplex formation, which is greater the less destabilizing the mismatches and the longer the repeats are.

Interestingly, we observed the impact of several single-stranded RNA repeats on the self-complementary structures of other repeats. For example, G(GCG)_3_G oligonucleotide forms a stable self-complementary duplex, but adding an equimolar amount of C(UGC)_3_U results in the formation of intermolecular duplex G(GCG)_3_G/C(UGC)_3_U of the stability of −15.35 kcal/mol. On the other hand, the mixing of G(GCG)_3_G with U(CUU)_3_C or U(CCU)_3_C did not disrupt self-complementary G(GCG)_3_G duplex (Figure 3B). We also observed that quadruplex structures formed by G(AGG)_3_A (Figure 3A) are disturbed in the presence of C(GCC)_3_G, U(CGU)_3_C, U(CUU)_3_C, and U(CCU)_3_C repeats. These four oligonucleotides form stable, non-self-complementary duplexes with the G(AGG)_3_A oligonucleotide and prevent the quadruplex formation (no transition at 295 nm during UV-melting). All four non-self-complementary duplexes melted in a two-state manner. Their stability was significantly higher than the G(AGG)_3_A quadruplex alone. The most stable duplex was the G(AGG)_3_A/U(CCU)_3_C duplex, for which we could obtain parameters in 100 mM NaCl buffer solution but not in 1 M NaCl (the melting temperature was higher than 93 °C). The least stable was G(AGG)_3_A/U(CGU)_3_C duplex.

The above results demonstrated that structure-free RNA repeats can bind to the ones that themselves fold into higher-order structures, unfolding them and resulting in the formation of new and more thermodynamically stable RNA structures. In native RNAs, these interactions are also dependent on the persistence of alternative structural motifs that are possible in a particular structural context.

### 2.4. In Vitro Structure Probing of RNAs with Multiple TNR Tracts

#### 2.4.1. Searching for the Exemplary RNAs

The thermodynamic parameters reported above for short RNA repeats reflect the potential of their mutual interactions, which is interesting, especially for tracts with no structure. We aimed to examine the relation between two selected TNR tracts coexisting in a single RNA in the natural sequence context, determining the secondary structure of the selected RNA fragments.

Analysis of trinucleotide repeats coexistence in mRNAs and ncRNAs confirmed the earlier reports and showed that CNG-type repeats, usually located in the 5′UTR and ORF, are dominant [10,26]. It identified only several transcripts containing more than one tract of rare repeats. The resulting transcripts were filtered by region of occurrence and distance between two tracts (Table 2). The distribution of the number of transcripts identified with four and six repeats of different types is shown on the heatmap (Figure 4). It shows a significant decrease in the number of RNAs found after the number of repeats increased by two. The vast majority of the results concerned the CNG repeats. Coexisting non-CNG repeats greater than six are extremely rare. To analyze the interaction between different repeat types, two RNAs for secondary structure probing were chosen: a fragment of 3′UTR *GABRA4* RNA and an ORF fragment of *CHIC1* RNA. The first RNA contains thermodynamically weak repeat types of (UUG)_10_ and (UAA)_14_; the second one contains thermodynamically stronger repeats (AGG)_4_ and (CGU)_4_, interleaved by AAG triplets. The selected RNAs represent two varied, thermodynamically interesting cases of RNA repeats’ interactions.

#### 2.4.2. Interaction of Unstructured Trinucleotide Repeat Tracts—The *GABRA4* 3′UTR

As we showed above in thermodynamic studies, the oligonucleotides consisting of UUG and UAA repeats do not form secondary structures by themselves but can interact with each other to form a duplex. By screening the transcripts for the coexistence of rare unstructured repeats, we found that the (UUG)_10_ and (UAA)_14_ repeats exist in all three variants of the *GABRA4* transcript in its 3′UTR region and are approximately 400 nucleotides apart. The structure–function relationship of *GABRA4* 3′UTR is unknown; however, the relative closeness of the mentioned repeats suggests that they may interact with each other, playing a potential role for the molecule. Its 3′UTR region consists of over 9300 nucleotides, which suggests a wide range of regulation of its expression.

We performed in silico structure prediction of the whole 3′UTR region of *GABRA4* (9325 nt) using the RNAFold online tool [27,28]. It showed that repeating sequences of UUG and UAA interact with each other, thus separating a small domain in this UTR (Appendix A). The 404-nucleotide long fragment of the 3′UTR (region of 3933–4337 nt of the whole mRNA) that makes up the domain was selected for detailed secondary structure chemical probing.

The chemical probing of the secondary structure was conducted with the SHAPE method [29] and chemical mapping with DMS, ketoxal, and CMCT [30,31,32]. All obtained data were introduced into the RNAstructure software [33] to predict the secondary structure of the *GABRA4* 3′UTR fragment. SHAPE data were introduced using “Read SHAPE reactivity—pseudo free energy” mode with slope 2.6 and intercept −0.8. From DMS, ketoxal, and CMCT mapping, only high reactive nucleotides were used as a chemical modification. The predicted model of secondary structure with the lowest free energy is highly structured (65% of the nucleotides are involved in base pairs). The 35% nucleotides are single-stranded and highly reactive with all chemical reagents (Figure 5).

The obtained secondary structure confirmed base pairing between the repeated tracts of UUG and UAA trinucleotides, which form a helix stem specifying the *GABRA4* 3′UTR domain. The entire domain consists of two major stem-loop structures forming subdomains 1 (region 41–295) and 2 (region 306–360). The subdomains are branching from the internal loop located next to the helix stem formed by the TNR tracts. The subdomain 1 stem includes seven internal loops of different sizes and six small hairpins. Three of the hairpins (regions 145–162, 169–188, 268–284) showed high reactivity to mapping agents, whereas the remaining three (regions 48–63, 65–78, 190–205) showed no reactivity. Within subdomain 1, high reactivity was also demonstrated by the internal loop regions 90–98 and 295–306. The subdomain 2 stem includes a single, seven-nucleotide-long poly-adenosine bulge, two uridine bulges, and a small, four-nucleotide-long loop (region 315–350), and each of the motifs was highly reactive. Regarding the base-pairing probability, it was the most structurally conservative region of the molecule. The region between the subdomains, which connects them, was also widely available for mapping reagents.

Secondary structure probability was performed using a partition function mode of the RNAstructure software. The results were visualized on the secondary structure with the lowest free energy. It was indicated that three hairpins show a very high probability of more than 99%, and one double-stranded region with internal loops (105–137 and 235–249) showed a probability higher than 95%. The TNR-containing helix structure showed a probability of more than 70%. The lower probability of base pairing in this region may suggest the potential lability of the helix and can be related to the dynamic function of this region. A three-dimensional model of the structure obtained in the RNAComposer program [34] showed that highly reactive nucleotides were often exposed outside the helix (Appendix A), which matches with their availability for mapping reagents.

3′UTRs are widely known as regulatory regions containing target sites for microRNAs. In the context of the huge 3′UTR of *GABRA4*, we have searched miRDB [35] for miRNA target sites. As a result, we have identified 148 potential microRNAs, with a significant target score above 80, from which 11 different ones are localized at 13 sites within the domain of our interest (Figure 6 and Appendix A). Four of them target the repeats (two for UUG and two for UAA repeats); however, from these four only the UAA-targeting ones was reported in the miRBase, as experimentally confirmed by NGS methods [36,37,38]. Nine miRNAs target subdomain 1 and three target subdomain 2. About half of the matched miRNAs bind to conserved regions of the predicted secondary structure. The most interesting site in this structure seems to be the loop of highly conservative hairpin 268–284. This nine-nucleotide-long loop presented very high reactivity during chemical mapping and is also a target site for three different miRNAs. Such high accessibility of the miRNA target sites suggests the possible active regulatory function of this region.

Our studies of the *GABRA4* 3’UTR fragment show that rare unstructured repeats are also able to mutually interact and fold into secondary structures, determining functional regions within an RNA molecule.

#### 2.4.3. Interaction of Purine-Rich Trinucleotide Repeat Tracts—The *CHIC1* ORF

*CHIC1*, also known as *BRX* (Brain X-linked gene), is a rare transcript preferentially expressed in the brain [39]. Its protein product is annotated through similarity to a membrane protein; however, its function is poorly understood. The gene is supposed to be the subject of the X-inactivation, as it lies within the XIC region (X-inactivation center). Within the human *CHIC1* RNA ORF sequence, several short tracts of AGG repeats are present that are separated by short tracts of AAG and CGU repeats. These repeats cover the codons for glutamate (GAG and GAA) and cysteine (UCG). The accumulation of AGG triplets suggests the formation of quadruplexes (G4 sequences) in the RNA structure. It is not clear whether the trinucleotide repeat pattern present in the transcript is significant at the transcript level due to its secondary and tertiary structure, or whether it is important for the encoded protein. The RNA structure of this region seems to be important; glutamate could be encoded by the GAA codon exclusively, but such repeats do not form a quadruplex, as evidenced by the above thermodynamic studies. G4 sequences in ORFs frequently encode low-complexity amino acid sequences, amino acid repeats, or short motifs [40,41], and contribute to translation-related processes such as elongation [42], ribosomal frameshift [43], no-go mRNA decay [44,45], and translational folding of newly synthesized proteins [46,47,48]. We chose a 170-nucleotide-long fragment of *CHIC1* ORF (region of 129–298 nt of the whole mRNA), containing cumulative repeats of AGG spaced by AAG, in the vicinity of (CGU)_4_ repeats, to define their structural relationships to one another. However, probing for the RNA secondary structure of this fragment failed in the first step due to the inability to obtain a cDNA. As predicted, the selected RNA fragment formed a quadruplex structure that effectively inhibited reverse transcription. The quadruplex formation was observed during UV-melting monitored at 295 nm, both in the presence of sodium and potassium ions; obtained curves were different, however. Due to the length of the analyzed *CHIC1* RNA fragment, the observed G4 structure is intramolecular, in contrast with intermolecular quadruplexes formed in the case of the earlier analyzed short oligonucleotides. Direct evidence of quadruplex formation is the melting curve for the guanosine-rich sequence which shows a “reversed” transition at 295 nm. Intramolecular quadruplexes (and hairpins) do not show Tm dependence on RNA concentration, but intermolecular structures (e.g., duplexes) do [49,50]. That is one of the reasons why the thermodynamic parameters of RNA are measured in several concentrations (usually from three to nine). The *CHIC1* RNA quadruplexes were also visualized on the non-denaturing gel stained using the NMM (N-Methylmesoporphyrin IX) reagent [51]. Two forms of the *CHIC1* RNA fragment have been observed, which suggests the existence of two differently migrating quadruplex structures (Figure 7B). The two independent online tools for G4 prediction, QGRS Mapper and G4PredictorTool, indicate four putative quadruplex-forming regions in the analyzed *CHIC1* ORF fragment.

Prediction of the secondary structure of this RNA fragment indicates that the CGU and AGG repeats do not interact directly with each other over their full length, but only at the ends of the tracts (Figure 7A). Unfortunately, it was impossible to reveal the exact manner of this interaction. The observed results indicate that G-quadruplex present in *CHIC1* ORF blocks mRNA-based processes, including the reverse transcription in our study. Further experiments are needed to explain the role of the quadruplex in this transcript, which might broaden the cognition perspective for the *CHIC1* function.

## 3. Materials and Methods

### 3.1. Oligonucleotide Synthesis

Oligoribonucleotides for UV-melting experiments, DNA primers for PCR reactions, and 5′-FAM labeled DNA primers for reverse transcription in SHAPE experiments were synthesized on a BioAutomation MerMade12 DNA/RNA synthesizer using β-cyanoethyl phosphoramidite chemistry as described previously [52], and commercially available phosphoramidites (ChemGenes (Wilmington, MA, USA); GenePharma (Shanghai, China)). The details of deprotection and purification of oligoribonucleotides were also described previously [52]. Thin-layer chromatography (TLC) purification of the oligonucleotides was carried out on Merck 60 F254 TLC plates with the mixture of 1-propanol/aqueous ammonia/water = 55:35:10 (v/v/v). After synthesis and purification, oligonucleotides were verified using mass spectroscopy (MALDI-ToF).

### 3.2. UV-Melting Experiments

The thermodynamic stability of structures formed by oligonucleotides composed of RNA repeats was analyzed in four buffers. Basic buffer was used to determine thermodynamic stability of the structure formed by RNA repeats and contained 100 mM sodium chloride, 20 mM sodium cacodylate, and 0.5 mM Na_2_EDTA, pH 7. The second buffer was used for the determination of the influence of the molecular crowding effect on oligonucleotides’ thermodynamic stability and also contained PEG 400 in a concentration of 80 g/L. If UV melting of the model repeat oligonucleotides in the buffers listed above indicated the formation of an unstable RNA structure, the oligonucleotides were analyzed in a buffer containing 1 M sodium chloride, 20 mM sodium cacodylate, and 0.5 mM Na_2_EDTA, pH 7. The physiological melting buffer contained 150 mM potassium chloride, 4 mM magnesium chloride, and 20 mM sodium cacodylate, pH 7. Oligonucleotide single-strand concentrations were calculated from the absorbance at 80 °C and single-strand extinction coefficients were approximated by a nearest-neighbor model. Absorbance vs. temperature melting curves was measured at 260 nm with a heating rate of 1 °C/min in the range of 4 to 90 °C on a JASCO V-650 spectrophotometer with a thermoprogrammer. Melting curves were analyzed and thermodynamic parameters were calculated from a two-state model with the program MeltWin 3.5. Statistical analysis of thermodynamic data was carried out using MeltWin 3.5 software. Based on results obtained for three to nine concentrations of analyzed oligonucleotides, thermodynamic parameters and melting temperatures were calculated.

### 3.3. Selection of RNAs for Structural Studies

To determine the possible interactions between various RNA repeat tracts, we searched the human RefSeq RNA database (NCBI) for natural RNAs containing more than one repeat tract. The selection of model RNAs for structural studies was based on the following criteria: (i) the presence of a minimum of two different repeat tracts, (ii) the minimum number of repeats within the tract is four, (iii) relatively close neighborhood of the different tracts (up to 400 nt) within one functional RNA region and (iv) at least one of the repeats is predicted not to form a secondary structure by itself. The inquired sequence contained ten repeats of trinucleotide. A tetranucleotide CCUG repeat was also included in the search for the study, due to the potential for the formation of structures and the connections with human muscle diseases [53]. In this case, an inquired sequence contained six repeats. The RefSeq RNA database was searched using megaBLAST with default algorithm parameters and excluding models (XM/XP). The results of the search were limited to mRNAs and ncRNAs and divided by repeat type and its region of occurrence in RNA.

### 3.4. Preparation of Target RNA Fragments

RNA fragments for structure probing were prepared by in vitro transcription from a DNA template prepared in a multistep PCR reaction from chemically synthesized oligonucleotides (Appendix A). The sequence of interest was preceded by a T7 polymerase promoter sequence and was used as a template for in vitro transcription with the MEGAshortscript™ Kit from Ambion/Life Technologies (Carlsbad, CA, USA) according to the manufacturer’s instructions. The correctness of obtained PCR products that served as matrices for in vitro transcription was verified each time by Sanger sequencing.

### 3.5. Structure PROBING for Target RNAs

#### 3.5.1. Chemical Mapping

For a single experiment, 2 pmol of RNA was used. The RNA was folded in a buffer containing 300 mM sodium chloride, 5 mM magnesium chloride, and 50 mM HEPES, pH 7.5 for 5 min at 65 °C, and slowly cooled down for 30 min. Four separate mapping reactions were conducted, each with different chemical reagents. The SHAPE method with NMIA (N-methylisatoic anhydride) was used to modify flexible 2′-hydroxyl of nucleotide residues. CMCT (1-cyclohexyl-3-(2-m-orpholinoethyl) carbodiimide metho-p-toluene sulfonate) alkylated the N3 of uridines not involved in base pairing and modified N1 of non-paired guanosine. Kethoxal alkylated the N1 and 2-amino group of non-paired guanosines, whereas DMS (dimethyl sulfate) was used to modify unpaired adenosines and cytidines. Depending on the reagent, RNA was treated with 8 mM NMIA for 60 min, 30 mM DMS for 60 min, 9.5 mM of CMCT for 30 min, or 6.7 mM of kethoxal for 20 min. All samples were incubated at 37 °C. A control reaction was performed in the same conditions without particular reagents. Mapping reactions were terminated by ethanol precipitation.

#### 3.5.2. Primer Extension and Mapping Data Analysis

Particular chemically mapped nucleotides were recognized by reverse transcription, starting from proper 5′-FAM labeled primers specific to tested RNAs (Appendix A). Some of the primers contained LNA-modified nucleotides that raise the stabilities of RNA–DNA duplexes, simultaneously increasing transcription efficiency [54]. A reverse transcription reaction was performed using SuperScript III First-Strand Synthesis system (Invitrogen (Waltham, MA, USA)) according to the manufacturer’s instructions, with some modifications as described previously [55]. Reactions were terminated by ethanol precipitation. For every probe, four ddNTP ladders were prepared. The final DNA product was separated by capillary electrophoresis using ABI 3130xl Genetic Sequencer. The readout was performed in the Laboratory of Molecular Biology Techniques at Adam Mickiewicz University in Poznan.

Chemically mapping data, such as ABI files, were analyzed with the Peak Scanner 1.0 Program (Applied Biosystems (Waltham, MA, USA)). Reactive nucleotides were recognized by comparing dideoxy sequencing ladders along with a mass marker. The first 2% of the highest peaks was excluded and the next 8% of the peak intensities was used to calculate the average value. To receive proper nucleotide reactivity, all peak intensities were divided by the average. Reactivities were normalized on a scale of 0 to 1.5, where 0 suggests not reactive site and 1.5 is highly reactive. Reactivities >0.7 are considered as strong, 0.3–0.7 as medium, and <0.3 as weak. All calculated data were used for secondary structure prediction. For all four mapping reagents, NMIA, DMS, CMCT, and kethoxal, at least three datasets were collected.

## 4. Conclusions

Trinucleotide repeats are among the most common in the transcriptome in terms of motif length. The presented study focused on the poorly described trinucleotide repeats, which are less frequent in the human transcriptome. In our work, we examined the stability and folding of 23 types (different sequences) of trinucleotide repeats, determining their interaction potential within their sequence as well as between different repeat types. The tested RNA repeats form duplexes, quadruplexes, or stay single stranded. Our data show, next to the CNG repeats type, CAA repeats can also self-fold into duplex structures. The presence of PEG, imitating molecular crowding in melting buffer, does not influence the stability of the RNA repeat structures. AU-rich trinucleotide repeats do not fold themselves and remain single stranded in the analyzed buffers, including crowding conditions. We observed their considerable potential for interaction with other repeat types, as the presence of the other partially complementary repeat-type RNA resulted in formation of the weak intermolecular duplex. Such intermolecular repeat interactions were also able to unfold AGG repeats-formed quadruplexes, and this happened in the presence of CGU, CCG, CUU, and CCU repeats. In the presence of magnesium and potassium ions, which bring the analyzed conditions closer to the physiological ones, duplexes formed by AU-rich trinucleotide repeats were destabilized up to 2 kcal/mol. We verified the interactions of the selected repeat tracts, observed in UV-melting experiments, by examining the secondary structure of two exemplary fragments of natural RNAs, in which two different repeat types coexist. Fragments of *GABRA4* and *CHIC1* RNAs, containing potentially interacting repeat tracts, were subjected to the chemical mapping of the secondary structure. The results obtained for the fragment of *GABRA4* RNA confirmed the formation of structures observed in thermodynamic studies of short model oligonucleotides. The in silico predictions performed for analyzed RNAs distinguish motifs formed by the repeats, which are consistent with the in vitro secondary structure probing results. *GABRA4* case showed that rare repeats consisting mainly of A and U residues can actively interact intramolecularly, forming a stem that separates a structural domain. The *CHIC1* case showed that the presence of a short CGU repeat tract was not able to unfold the G4 structure intramolecularly, which, however, was probably more complex, as it was formed by four short, discontinuous AGG repeat tracts.

Trinucleotide repeats in a number above six, which themselves do not have secondary structures and remain single stranded, are very rare in the human transcriptome. Even more rare is the coexistence of such repeats within one transcript. CNG-type repeats, usually located in the 5′UTR and ORF, are dominant. However, we showed on the *GABRA4* example that duplexes formed from different, potentially weak RNA repeats might be structurally important for the molecule function and regulation. RNAs found with more than six repeats usually contain repeat types that themselves easily fold into secondary structures and localize in ORFs. As the RNA sequence context makes up the RNA structure, it is natural that self-unstructured repeats usually interact somewhere in the context in the most optimal manner. Our question in the case of *GABRA4* was if the interaction of the repeat tracts UUG and AAU will be more thermodynamically favorable than the other interactions in their sequence context. The obtained experimental structure of the *GABRA4* fragment confirmed the superiority of this interaction and its shaping role in the structure of this region.

## Figures and Tables

**Figure 1 ijms-23-05850-f001:**
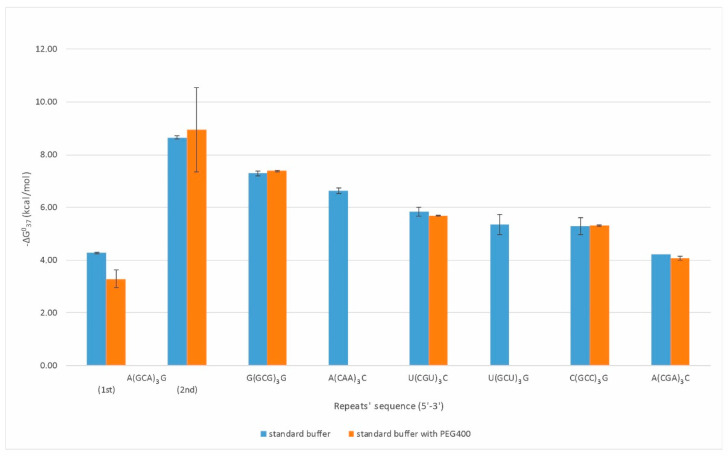
Thermodynamic stability of self-duplexing RNA trinucleotide repeats.

**Figure 2 ijms-23-05850-f002:**
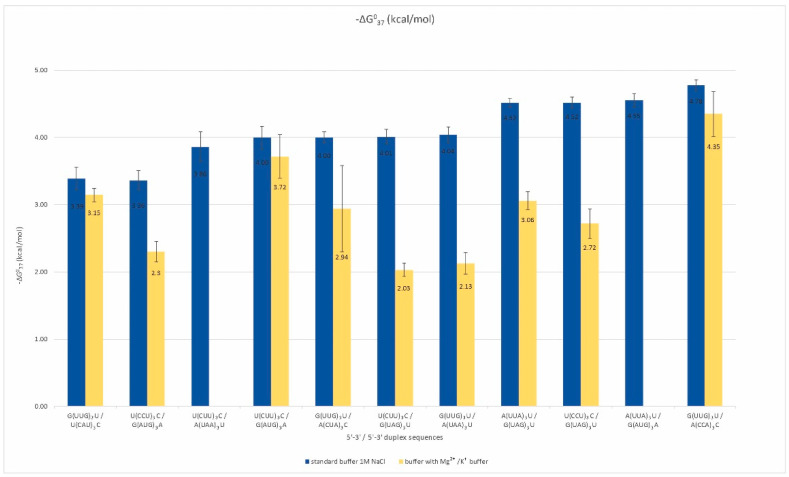
Thermodynamic stability of inter-strand duplexes of non-self-duplexing RNA repeats.

**Figure 3 ijms-23-05850-f003:**
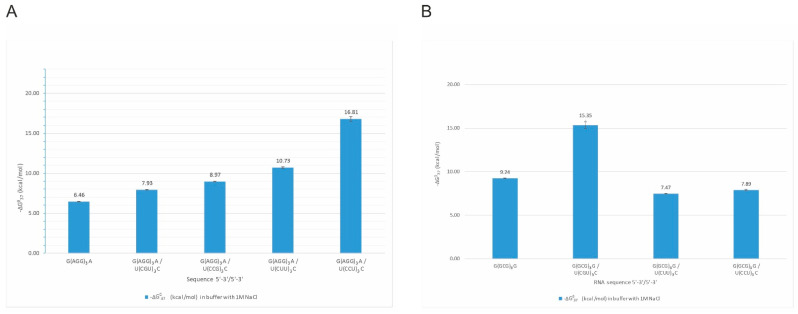
Thermodynamic stability of intermolecular duplexes formed by G(AGG)_3_A repeats (**A**) and G(GCG)_3_G repeats (**B**).

**Figure 4 ijms-23-05850-f004:**
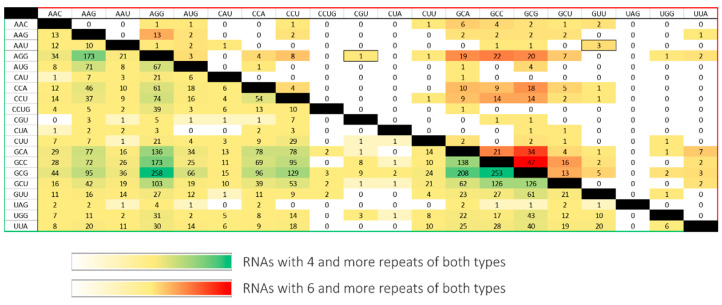
RNA repeats coexistence in a single transcript. The values on the crossing of particular repeats type row and column represent the number of RNAs found with the four (yellow–green scale) and the six (yellow–red scale) repeat number.

**Figure 5 ijms-23-05850-f005:**
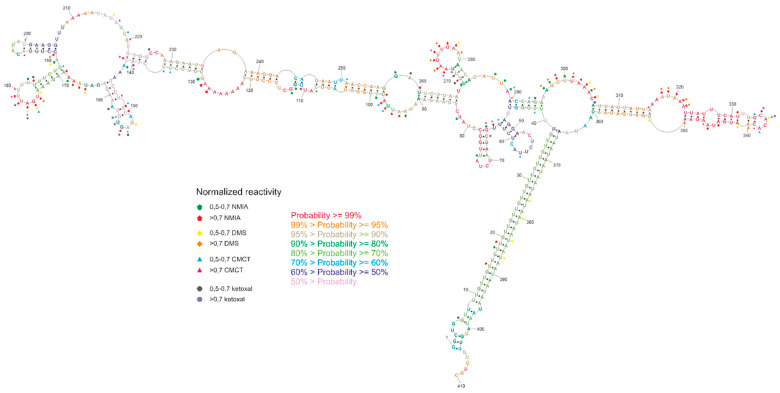
Experimental secondary structure of *GABRA4* 3′UTR fragment, including normalized nucleotide reactivity’s and base-pairing probability.

**Figure 6 ijms-23-05850-f006:**
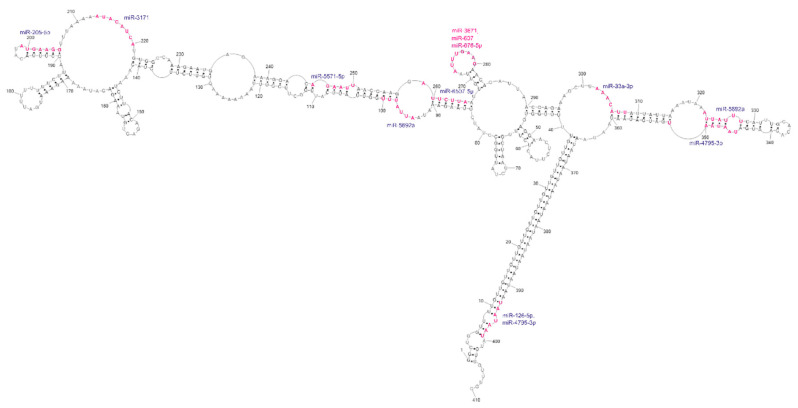
Analysis of potential miRNA target site regions within the *GABRA4* 3′UTR fragment. The sites of interactions significantly overlap with the reactive sites determined by chemical structure probing.

**Figure 7 ijms-23-05850-f007:**
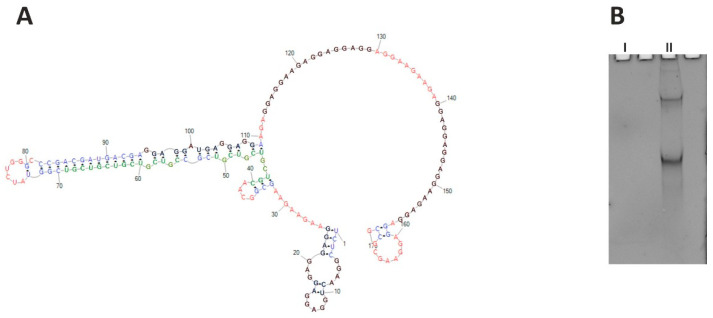
(**A**) Structure prediction for *CHIC1* ORF RNA fragment (RNAstructure). Putative quadruplexes are marked black, and CGU repeats are marked green; red—single stranded regions, blue—duplex regions. (**B**) Native 8% polyacrylamide gel stained with NMM, showing two *CHIC1* isoforms containing quadruplex structures (II). Control RNA (I) of similar length is G4-free and therefore invisible.

**Table 1 ijms-23-05850-t001:** RNA repeats potential for folding into different secondary structures.

Unstructured	Duplex	Quadruplex
A(UUA)_3_U	A(CAA)_3_C	G(AGG)_3_A
G(AAG)_3_A	A(CGA)_3_C	G(UGG)_3_U
G(UUG)_3_U	U(GCU)_3_G	
A(GAA)_3_G	U(CGU)_3_C	
A(UAA)_3_U	C(GCC)_3_G	
U(CUU)_3_C	G(GCG)_3_G	
U(CCU)_3_C	A(GCA)_3_G	
A(CCA)_3_C	G(CCUG)_3_C	
A(CUA)_3_C		
U(CAU)_3_C		
G(UAG)_3_U		
G(AUG)_3_A		
U(AUUCU)_3_A		

**Table 2 ijms-23-05850-t002:** Top results of megaBLAST search for RNAs with multiple TNR tracts in human transcriptome. Records marked in red were chosen for detailed analysis of RNA secondary structure to follow trinucleotide repeat tracts interactions. Records marked with an asterix contain discontinuous tracts of repeats.

	Gene ID	Variant	Repeat Type	Repeats Number	Transcript Region	Region Lenght	Distance between Repeats	Codon Type
**Non-CNG type**	GABRA4	NM_000809	UUG	10	3′UTR	9325	318	-
			UAA	14	3′UTR			-
	ALG13	NM_001099922	CCA	14	ORF	3414		Pro
			CCU	13	ORF		0	Pro
	HRC	NM_002152	AGG	8	ORF	2100		Asp
			AUG	14	ORF		138	Glu
	ARID3A	NM_005224	UGG	10	5′UTR	292		
			AGG	6	ORF	1782	434	Glu
**CNG type**	ZFHX3	NM_001164766	GCC	10	5′UTR	515		
			AAC	9	ORF	8370	2642	
			GCA	7	ORF		4369	Gly
			GCG	7	ORF		947	
	MAFA	NM_201589	GCG	5	ORF	1062		His
			CCA	10	ORF		352	Gly
	ATXN8OS	NR_002717	CUA	10	ncRNA	1472		
			GCU	14			0	
	RPS6KA6	NM_014496	GCG	11	5′UTR	300		
			CUA	12	3′UTR	5927	3637	
	AR	NM_000044	GCA	23	ORF	2763		Gln
			GCG	17	ORF		1130	Gly
	VEZF1	NM_001330393	GCC	6	5′UTR	382		
			GCA	13	ORF	1539	1298	Gln
	HTT	NM_002111	GCA	21	ORF	9435		Gln
			GCC	7	ORF		3	Pro
	AAK1	NM_014911	GCA	6	ORF	2886		Gln
			GUU	10	3′UTR	17,872	12,815	
	POU4F2	NM_004575	GCG	11	ORF	1230		Gly
			CCA	6	ORF		331	His
	HMGB3	NM_001301228	GCC	17	5′UTR	262		
			AGG	7	ORF	603	663	Glu
	** SKIDA1	NM_207371	GCC	10, 8	ORF	2727		Ala
			GCG	8, 4	ORF		174	Ala
			CCA	7, 6	ORF		65	His
			AGG	5, 4	ORF		188	Glu
	** ZSWIM6	NM_020928	GCC	10, 4	ORF	3648		Ala
			GCG	6, 6, 6, 6	ORF		44	Gly
	** KCNMA1	NM_001014797	GCG	6, 5, 7	ORF	3549		Gly
			CCU	7	ORF		60	Ser
			CUU	4	ORF		0	Ser
	** CHIC1	NM_001039840	CGU	4, 5	ORF	675		Ser
			AGG	4, 4, 3	ORF		23	Glu

**: The repeat tracts are discontinuous.

## Data Availability

Not applicable.

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
