# Peer review of "A Structural Potential of Rare Trinucleotide Repeat Tracts in RNA"

_ijms, 2022, doi:10.3390/ijms23105850_

Round 1

Reviewer 1 Report

The present manuscript is an attempt to identify the potential contribution of RNA trinucleotide repeats to the structure of RNAs that contain these repeats. These was first done by in-vitro melting experiments, followed by in-silico analysis of the putative structure of two mRNAs.

The study is overall interesting, as in-silico predictions match the experimental outcomes, although not systematically. 

The manuscript is readable and does not require further experimentation, however the quality of the written English is at time low. We provided some editing all along to improve the overall quality of the manuscript. Likewise, the authors often did not include the published references in the manuscript. Finally, the numbering of the figures will need to be updated as some tables have been wrongly labelled as Figures.

All required editing is available on the attached PDF file. Once all corrections and amendments have been made, the manuscript is acceptable for publication. 

Author Response

Dear Reviewer 1,

Thank you a lot for your comments and suggestions.

I corrected language errors, changed the captions and numbering of tables and figures, and added published references, according to your suggestions from the attached pdf file.  At present, the manuscript contains 55 references versus 39 in the original submission. I also rewrote/added several sentences (lines 56-58, 213-218,422-427) to make the text more understandable. 

I hope that the changes that I made are satisfactory and now the manuscript meets the requirements for publication.
Kind Regards
Dorota Magner

Reviewer 2 Report

This is a good paper that addresses the problem of the formation of various types of structure in RNA sequence by thermodynamic studies of selected repeats. The results will be useful both to the RNA experimental community and biologists/biophysicists and to theoreticians who attempt at the construction of the force fields for RNA. The experiments were performed and analyzed correctly, the conclusions are sound. The paper is well and concisely written, it reads easily. I have two minor suggestions:

  1. For non-specialists in the field, it would be useful to indicate how the experiments performed enabled the authors to distinguish the formation of quadruplexes/hairpins/intertrand quadruplexes from that of regular intrastrand duplexes.
  2. The following sentence in the Introduction (lines 55 and 56) seems to be unclear: "On the other side,
    many RNA repeats do not form any particular structure, their presence in the human ge-
    nome may suggest some biological functions, however."

Author Response

Dear Reviewer 2,

Thank you for your comments. 

Point 1.
In lines 422-427 I added sentences that clarify the way of interpretation of melting results for inter- and intramolecular RNA structures:
"Direct evidence of quadruplex formation is the melting curve for the guanosine-rich sequence which shows a “reversed” transition at 295 nm. Intramolecular quadruplexes (also hairpins) do not show Tm dependence on RNA concentration, but intermolecular structures (e.g. duplexes) do [49,50]. That is one of the reasons why the thermodynamic parameters of RNA are measured in several concentrations (usually from three to nine). " 
I added also references to this topic.

Point 2.

I replaced the sentence from lines 55-56 with the following:
"On the other side, many RNA repeats do not form any particular structure, but their presence in the human genome may suggest some yet undetermined biological functions." 

I have also corrected some linguistic errors, added references, and changed Figure 1 into Table 1 and Figure 5 into Table2. Therefore, the numeration of the rest figures has also changed.

I hope that the changes I have made are satisfactory and now the manuscript meets the requirements for publication.

Kind Regards

Dorota Magner